# Study of *Salmonella* spp. from Cage Papers Belonging to Pet Birds in an Argentinean Canary Breeder Championship

**DOI:** 10.3390/ani14081207

**Published:** 2024-04-17

**Authors:** Dante J. Bueno, Francisco I. Rodríguez, Luciana C. Machado, Mario A. Soria, Francisco Procura, Silvana C. Gómez, Teresa M. Hoffmann, Andrea Alcain, María I. Caffer, Juan D. Latorre, Javier O. Quintar

**Affiliations:** 1Instituto Nacional de Tecnología Agropecuaria EEA Concepción del Uruguay, Ruta Provincial 39 Km 143.5, Concepción del Uruguay E3260, Entre Ríos, Argentina; mas_1975@hotmail.com; 2Facultad de Ciencia y Tecnología sede Basavilbaso, Universidad Autónoma de Entre Ríos, Barón Hirsch Nº 175, Basavilbaso E3170, Entre Ríos, Argentina; 3Agencia Santafecina de Seguridad Alimentaria, Francia, Santa Fe S3000, Santa Fe, Argentina; francisco_rodriguez_fir@hotmail.com; 4BBR Industries Argentina, Calle 70, La Plata B1904BHQ, Buenos Aires, Argentina; machadolucianac@gmail.com; 5Facultad de Bromatología, Universidad Nacional de Entre Ríos, Pte. Perón 1154, Gualeguaychú E2820, Entre Ríos, Argentina; franciscoprocura@hotmail.com; 6Facultad de Ciencia y Tecnología, Universidad Autónoma de Entre Ríos, 25 de Mayo 353, Concepción del Uruguay E3260, Entre Ríos, Argentina; gomezsilvanac08@gmail.com; 7Consejo Nacional de Investigaciones Científicas y Técnicas (CONICET), Departamento Avicultura, Instituto Nacional de Tecnología Agropecuaria EEA Concepción del Uruguay, Ruta Provincial 39 Km 143.5, Concepción del Uruguay E3170, Entre Ríos, Argentina; magalihoffmann@gmail.com; 8Servicio Enterobacterias, Instituto Nacional de Enfermedades Infecciosas (I.N.E.I.), Dr. “Carlos G. Malbrán”, Av. Vélez Sarsfield 583, Ciudad Autónoma de Buenos Aires C1282AFF, Argentina; aalcain@anlis.gov.ar (A.A.); caffermi@fibertel.com.ar (M.I.C.); 9Department of Poultry Science, University of Arkansas Agricultural Experiment Station, Fayetteville, AR 72701, USA; jl115@uark.edu; 10Los Twity, Estrada 575, Concepción del Uruguay E3260, Entre Ríos, Argentina; javierquintar@cladan.com.ar

**Keywords:** canary, *Salmonella*, bird fairs, Argentina, antibiotic, disinfectant

## Abstract

**Simple Summary:**

This study was conducted to estimate the rate of *Salmonella* spp. isolation from cage papers, located in the bottom of canaries’ cages and those of other exotic pet birds, in the 2015 Argentinean canary breeder championship. Furthermore, we determined the antimicrobial resistance profile of the isolates for antibiotics and commercial disinfectants. One pool of 258 cage paper pools was positive for *Salmonella* spp. (0.4%). Two strains of *Salmonella* serotype Glostrup were isolated, which were only resistant to sulfonamides and erythromycin and sensitive to alkali-based product PL301 AS. Although the rate of *Salmonella* spp. isolation from cage papers in an Argentinean canary breeder championship is low, this is the first study conducted in Argentina on *Salmonella* spp. isolation from these materials of pet birds. However, *Salmonella* ser. Glostrup isolated strains could be a source of human *Salmonella* outbreaks, and they show high resistance to disinfecting products.

**Abstract:**

Birds, including canaries and other birds, have become increasingly popular as pets. Bird fairs, where breeders gather and show their production in a championship setting, present a setting for possible *Salmonella* spp. contamination and transmission. Therefore, this study estimated the rate of *Salmonella* spp. isolation from cage papers, located in the bottom of cages of exotic pet birds, including canaries. Collected *Salmonella* isolates were used to determine the antimicrobial resistance profile to 52 antibiotics and 17 commercial disinfectants, based on pure or a mixture of acids, alcohols, aldehydes, alkalis, halogens, peroxygen, and quaternary ammonium compounds. The samples consisted of 774 cage papers taken in the 2015 Argentinean canary breeder championship, pooling three cage papers into one sterile sampling bag. Only one pool of the cage papers was positive for *Salmonella* spp. (0.4%), which belonged to the sample from three frill canary cages. Two strains of *Salmonella* serotype Glostrup were isolated, which were only resistant to sulfonamides and erythromycin and sensitive to alkali-based product PL301 AS. Although the rate of *Salmonella* spp. isolation from cage papers in an Argentinean canary breeder championship is low, it should not be discounted because *Salmonella* ser. Glostrup can be a source of human *Salmonella* outbreaks and they show high resistance to disinfecting products.

## 1. Introduction

Different species of birds have become increasingly popular as pets. Of all pet birds, *Serinus canarius* (canary) is considered a domesticated species [1]. There are three groups of canaries: song, color, and form. Song canaries include Harzer (Germany), Malinois (Belgium), Timbrado (Spain), and American singers (USA). Color canaries are divided into two groups: melanin (black, brown, agate, and isabel birds) and lipochrome (white, red, and yellow). Form canaries are a diverse group, distributed between frill, type, shape, crested (and crest-bred) birds, and feather-pattern birds [2].

*Salmonella enterica* is a diverse bacterial species, currently divided into six subspecies and more than 2500 serovars [3]. Many serotypes are important pathogens in humans and animals with varying levels of host specificity [4]. Avian species without ceca, like the canary, or with involute ceca appear to be more susceptible to *Salmonella* infections than birds with fully functioning ceca [5]. This bacterium produces one of the most important bacterial diseases in canaries [2].

Whether or not to treat *Salmonella* infections in companion birds is controversial. However, due to public health hazards, antibiotic treatment is usually recommended for clinically affected birds and companion birds that are identified as carriers [5]. A culture and antibiotic sensitivity test result is performed to determine the most effective antibiotic to administer [2,5]. There are few studies of *Salmonella* susceptibility to different antimicrobial agents in canaries [6,7,8,9]. Furthermore, chemical disinfectants are often the first line of defense against pathogens, and these antimicrobial products vary in their ability to destroy microorganisms. Reducing the amount of pathogens on surfaces lowers the chance of exposure and illness risks, safeguarding the health of animals and people who interact with them [10]. However, guidelines for disinfectant selection and usage often are defined less strictly. Although different studies are available on the in vitro efficacy testing of commercial disinfectants against *Salmonella* serovars with different contamination methods [11,12,13,14], they do not use strains isolated from canaries or their environment.

There are competitions and exhibitions of canariculture at local, regional, national, international, and global scales [15]. In Argentina, the Argentinean Federation of Canary Breeders organizes a championship every year, which is held over a few days, and exotic pet bird breeders present their bird pets, especially different types of canaries. Although the National Poultry Health Plan includes some *Salmonella* serovars in the control plan for poultry in Argentina, utilizing culture methods in the laboratory procedure [16], canaries and other pet birds are usually not vaccinated or monitored for diseases/microorganisms. These birds can be clinically affected by *Salmonella* serovars or subclinical carriers and can serve as a reservoir for an aviary [5,7]. Consequently, these infected birds can contaminate the environment through cage papers, feces, or feathers. They may also transmit infections to humans and other birds either indirectly or directly from handling [17]. Paper and paper products are materials often used for bedding in bird cages because it is very easy to change, cheaper than other materials, relatively inexpensive, and allows for the visual monitoring of droppings. However, it is a rich environment for bacterial growth because of the contamination of dropped food, spilled water, and bird feces [18]. For that reason, this study was conducted to (1) study the rate of *Salmonella* spp. isolation from papers, located in the bottom of the cage of canaries and other exotic pet birds during an Argentinean canary breeder championship; and (2) determine the antibiotic and disinfectant resistance profile of the isolated bacteria.

## 2. Materials and Methods

### 2.1. Sample Collection

Approximately 13,000 canaries, 73 canary hybrids and 117 other exotic pet birds from 44 associations, that belonged to the Argentinean Federation of Canary Breeders arrived at La Rural property, Buenos Aires City, Argentina, on 30 June 2015 to participate in the 65th Argentine Canary Breeder Championship [19,20]. Upon arrival, the birds received electrolytes in their drinking water to decrease their susceptibility to potential pathogens due to stress during transport. During the entire championship, the birds that presented clinical signs were monitored and medicated if needed.

The cage white paper (one/cage) was placed in the bottom of each cage (30 × 23 × 25 cm) on the morning of 31 June 2015. A total of 774 cage papers, used as bedding, were randomly removed, using disinfected gloves, from the cages of birds on 2 July 2015. This also included the bird feces and the remainder of their food, feathers, and water that was on the paper. The Argentinean Federation of Canary Breeders approved this sampling work. Three cage papers were pooled and put into one sterile sampling bag. Each sterile bag was labeled with cage numbers to document which bird’s cage paper was being tested. Two hundred fifty-eight sterile bags, which included cage papers from 744 canaries, 15 canary hybrids, and 15 exotic pet birds (Table 1), were transferred to the INTA Bacteriology Laboratory of Poultry Department (Concepción del Uruguay, Entre Ríos, Argentina) in ice chests for *Salmonella* spp. isolation.

### 2.2. Salmonella *spp.* Isolation and Identification

In the laboratory, 100 mL of buffered peptone water (Acumedia-Neogen, Lansing, MI, USA) was added to each sterile plastic bag. The mixture was incubated at 35 ± 2 °C for 18–24 h. One milliliter of incubated broth was transferred to 10 mL of tetrathionate broth base (Acumedia-Neogen), containing 20 mL/L of iodine potassium iodide solution (6 g of iodine; 5 g of potassium iodide; 20 mL of demineralized water), brilliant green 0.1% (Sigma-Aldrich Chemie GmbH, Steinheim, Germany), and 40 mg/mL of novobiocin (Sigma-Aldrich Chemie GmbH), and incubated at 35 ± 2 °C for 18–24 h. After that, samples were streaked on two selective and differential media, xylose lysine desoxicholate agar (XLD; Difco^TM^, Sparks, MD, USA), and EF-18 agar (Acumedia-Neogen) and incubated at 35 ± 2 °C for 18–24 h.

Two or more presumed *Salmonella* colonies on each selective–differential agar plate were picked and biochemically confirmed using triple-sugar iron agar (Acumedia-Neogen), lysine iron agar (Acumedia-Neogen), ortho-nitrophenyl-β-galactoside (Laboratorios Britania, Buenos Aires, Argentina), Simmons citrate (Merck, Darmstadt, Germany), sulfide indole motility medium (Merck), Jordan’s tartrate agar, phenylalanine agar (Hi-Media, Mumbai, India), and urea agar (Laboratorios Britania). If no bacterial colonies comparable with *Salmonella* spp. in a selective–differential agar plate were present, two atypical *Salmonella* spp. colonies were picked and tested using the same biochemical tests listed above. All *Salmonella* isolations were preserved on nutritive (Merck) slant agar until serotyping, which was carried out according to the White–Kauffmann–Le Minor scheme, with somatic and flagellar antigens [21].

### 2.3. Antibiotic Susceptibility Test

The antibiotic susceptibility test was performed by the standard disk diffusion method in Mueller–Hinton agar (Difco, Sparks, MD, USA) and the results were expressed as susceptible, intermediate, or resistant, according to the Clinical and Laboratory Standards Institute [22,23,24]. The strains were screened for resistance to 52 antibiotics belonging to 14 antibacterial classes (Table 2). *Pseudomonas* spp. breakpoints were adopted from the Clinical and Laboratory Standards Institute [22] for colistin, as no Enterobacterales criteria for disc diffusion antibiotic resistance assay has been defined either by this Institute or the European Committee on Antimicrobial Susceptibility Testing. The zone diameter breakpoint used for fosfomycin/tylosin was the same as for fosfomycin. *Escherichia coli* ATCC 25922 was used as control strain. An isolate was classified as multidrug-resistant (MDR) if it resulted as non-sensitive to at least one agent in three or more antibacterial classes [25]. Multiple antibiotic resistance index (MARI) for each resistance pattern was calculated too, by employing the formula proposed by Krumperman [26]: MARI = number of resistance antibiotics/total number of antibiotics tested. Isolates classified as intermediate on the basis of inhibition zone were considered as sensitive for the MARI [27]. Additionally, we evaluated two more methods for testing colistin, a commercial spot agar (ColTest, Laboratorios Britania) that uses a proprietary concentration of colistin (3 µg/mL), using the same procedure as the one described by Pasteran et al. [28], and a broth disk elution [23].

### 2.4. Testing of Disinfectants

The strains were in vitro screened in nine concentrations of 17 commercial disinfectants used in animal production, based on pure or mixture of acids, alcohols, aldehydes, alkalis, halogens, peroxygen, and quaternary ammonium compounds (Table 3). These concentrations included those recommended by the manufacturers. The dilutions of the products were prepared with sterile deionized water on the day of the test. The Mueller–Hinton agar (Laboratorios Britania) plates were inoculated by dipping a sterile swab into ~1.5 × 10^8^ CFU/mL (turbidity visually comparable to that of the 0.5 McFarland turbidity standard) *Salmonella* spp. strain suspension in tryptic soy broth (Neogen, USA) and streaking it across the surface of the agar in three directions. The plates were dried at ambient temperature for 15 min before applying 15 μL of each disinfectant dilution. Each plate was inoculated with 3 doses of one product and after 30–45 min it was incubated at 37 ± 1 °C for 18–24 h. A control was included in which the disinfectant was substituted by sterile deionized water. The minimum inhibitory dose (MID) was considered as the lowest concentration of a disinfectant necessary to inhibit visible growth (the diameter of the zone showing no obvious, visible growth, ≥5 mm, measured in two directions by a ruler) of the strain in the plate. The strains were reported as sensitive and intermediate to each disinfectant tested when the MID was less than or equal to the doses recommended by the manufacturer, respectively. The strains were considered resistant when the MID was greater than the maximum dose recommended by the manufacturer. The disinfectant doses recommended for each product by manufacturers are detailed in Table 3. Furthermore, a mixture of 15 μL of a mixture of 1% PL 308 DA (Proar Pilar, Fatima, Argentina) with 1% Profoam (Diversey), concentrations as it is used in the poultry industry, was applied on the plate with the strain as described above.

## 3. Results

Out of 258 samples, which included three cage papers each, only one pool was positive for *Salmonella* spp. (0.4%). From 1032 bacterial colonies taken from agar plates, two strains of *Salmonella* ser. Glostrup were isolated from EF-18 agar. These belonged to a pool of cage papers from three frill (Gibber italicus, Frisé Suisse and Fiorino) canary cages, from three different canary breeders.

*Salmonella* ser. Golstrup strains were not MDR strains and the MARI for these strains was 0.04 (2/52). They were resistant to sulfonamides and erythromycin and intermediate to cefazolin and ciprofloxacin. One strain was intermediate to ceftaroline and neomycin. The other strain was sensitive to these two antibiotics (Table 4). Both strains were sensitive to the rest of the antibiotics tested and to colistin by the standard disk diffusion method in Mueller–Hinton agar, CBDE (1 µg/mL), and ColTest.

In reference to disinfectant resistance, both strains were resistant to the mixture of two disinfectants (1% PL 308 DA with 1% Profoam). However, *S*. ser. Golstrup strains isolated showed some differences from the different disinfectants tested (Table 5). Based on the recommended doses for disinfection, both strains were only sensitive to the alkali-based product PL301 AS. One strain was resistant to 13 commercial products and the other showed resistance to eight disinfectants. The last one was sensitive to nine commercial products.

## 4. Discussion

In this study, the frequency of *Salmonella* spp. isolation serovars from cage papers at a canary breeder championship was 0.4%. There have been several studies on the prevalence of *Salmonella* infection among pet birds kept in cages, which is between 0.6% and 18.1% for canaries [7,8,9,29]. However, the samples generally included cloacal swabs, freshly dropped feces, and/or carcasses in these studies, but no cage papers like in our study.

*Salmonella* ser. Glostrup was the only serovar isolated from canary cage papers in the present study. These cage papers belonged to three different canary breeders, and we did not know if this serotype would come from one, two, or three cages. Different studies report that *Salmonella* serovar Typhimurium is the most frequently isolated serotype followed by *Salmonella* ser. Enteritidis in canaries [6,7,8,29,30]. Our study did not recover any *Salmonella* ser. Typhimurium (Table 4). Furthermore, there is not any information about *S.* ser. Glostrup in canaries. However, this serovar was isolated from poultry slaughterhouses [31], water from poultry facilities [32], captive reptiles [33], and human fecal samples [34]. Furthermore, it was reported that the family dog could be implicated as source of a human *S.* ser. Glostrup outbreak [35]. Therefore, cage papers from pet animals can represent a potential risk to other birds and humans.

In our study, *Salmonella* strains were sensitive to 46 antibiotics tested and only resistant to sulfonamides and erythromycin (Table 4). There were not any MDR strains, and the MARI was below 0.21, which is considered the lowest value to be considered high-risk [26]. It is known that Gram-negative bacilli, such as *Salmonella* spp., are usually intrinsically resistant to erythromycin, a macrolide antibiotic [36,37]. Rahmani et al. [7] reported that all *Salmonella* isolates from pet birds in Tehran, Iran, were susceptible to danofloxacin, norfloxacin, levofloxacin, amikacin, gentamicin, and tobramycin. Resistance to other antibacterial agents was variable and ranged from 0 to 57.9%. Although most of the strains were not completely serotyped in this study, six of them belonged to serogroup C like *S*. ser. Glostrup. Furthermore, although erythromycin was not tested in these studies, strains of this serovar, isolated from a duck carcass and human fecal samples, were susceptible to all of the antibiotics tested [31] or only resistant to gentamicin [34], respectively.

It is considered that the disk diffusion test, commonly used in clinical laboratories, is not reliable for measurements of in vitro colistin resistance because large molecular weight antibiotics, such as polymyxins, diffuse slowly into agar, resulting in small differences in the size of inhibition zones between susceptible and non-susceptible isolates [38]. However, *S.* ser. Glostrup strains were sensitive to colistin by three methodologies in our study: the standard disk diffusion method in Mueller–Hinton agar, CBDE, and ColTest. Recently, Pasteran et al. [28] reported that ColTest showed 97.9% categorical agreement for Enterobacterales with standard broth microdilution.

The selection of an appropriate disinfectant for an animal facility is a complex, multifactorial decision requiring consideration of spectrum of activity, human safety, environmental safety, and behavioral effects on animals [39]. Different reports showed that the ability of disinfectants to eliminate *Salmonella* spp. is influenced by the type of disinfectant chosen and its concentration [11,12]. In this way, we found some differences in disinfectant resistance profiles in the two *S*. ser Golstrup tested. Based on the maximum dose recommended by the manufacturer, both strains were only sensitive to product PL301 AS. Most of the commercial disinfectants that we tested were composed of a mix of substances. One strain was resistant to 13 commercial products and the other showed resistance to eight disinfectants. Therefore, in many cases, the recommended concentration of the products will not be efficient in eliminating these strains. Furthermore, although bleach (sodium hypochlorite) is the standard recommendation made by most Environmental Health and Safety departments [40,41], in our study (Table 5), no *Salmonella* strains were growth-inhibited when the recommended concentration of the commercial product End Bac 2, which contains sodium hypochlorite as the main component, was used.

## 5. Conclusions

This is the first study conducted in Argentina on *Salmonella* spp. isolation from the cage papers of pet birds. Although the rate of *Salmonella* isolation spp. from cage papers, located in the bottom of the cages of canaries and other exotic pet birds situated in an Argentinean canary breeder championship, is low, it should not be discounted since *S.* ser. Glostrup can be a source of human *Salmonella* outbreak and it shows high resistance to disinfect products (PL308 DA, Ucarsan 420, End Bac 2, GLUTASEPT, Cid 20, Pramolipa, Germatron, Aminol 50). On the other hand, multi-resistance is not a problem in the isolated strains, because they were only resistant to erythromycin and sulfonamides, and multi-resistance has not been a problem in these strains until now. Additional research to generate a larger number of isolates is necessary to adequately evaluate the prevalence of antimicrobial resistance among *Salmonella* isolated from cage papers. Finally, cage paper should be considered as a sample to add in the study related to diseases in pet birds, because these materials also can contain the feces of birds, feathers, food, and water, which can be a source of *Salmonella* infecting birds and humans.

## Figures and Tables

**Table 1 animals-14-01207-t001:** Pet birds whose cage papers were sampled in the 65th Argentine canary breeder championship for *Salmonella* spp. isolation.

Pet Bird	Name/Type		Number of Cage Papers Sampled
Canaries	Group	Color	607
		Form	137
Canary hybrids		Hooded Siskin (*Carduelis magellanica*) × canary	7
		European Goldfinch (*Carduelis carduelis*) × canary	3
		Eurasian Siskin (*Carduelis spinus*) × canary	2
		House Finch (*Carpodacus mexicanus*) × canary	2
		European Greenfinch (*Chloris chloris*) × canary	1
Other pet birds		Zebra finch (*Taeniopygia guttata*)	6
		Fischer’s lovebird (*Agapornis fischeri*)	3
		Cut-throat finch (*Amadina fasciata)*	2
		Rosy-faced lovebird *(Agapornis roseicollis)*	2
		Red-headed finch (*Amadina erythrocephala)*	1
		Cockatiel (*Nymphicus hollandicus*)	1

**Table 2 animals-14-01207-t002:** Antimicrobial agents and disc concentrations used for antibiotic susceptibility test over *Salmonella* spp. strains.

Antimicrobial Class	Antimicrobial Agent	Concentration (µg/disc)
Penicillins	Amoxicillin ^a^	10
	Ampicillin ^a^	10
	Piperacillin ^b^	100
Β-lactams combination agents	Amoxicillin–clavulanate ^a^	20–10
	Ampicillin–sulbactam ^c^	10–10
	Ceftazidime–avibactam ^a^	10–4
	Ceftolozane–tazobactam ^b^	30–10
	Piperacillin–tazobactam ^c^	100–10
Cephems	Cefazolin ^b^	30
	Cephalothin ^c^	30
	Cefaclor ^b^	30
	Cefoxitin ^a^	30
	Cefotaxime ^a^	30
	Ceftazidime ^a^	30
	Cefpodoxime ^a^	10
	Cefixime ^a^	5
	Ceftibuten ^b^	30
	Ceftiofur ^a^	30
	Cefepime ^c^	30
	Ceftaroline ^b^	30
Monobactams	Aztreonam ^c^	30
Penems	Doripenem ^b^	10
	Ertapenem ^b^	10
	Imipenem ^a^	10
	Meropenem ^c^	10
Aminoglycosides	Streptomycin ^b^	10
	Kanamycin ^a^	30
	Amikacin ^a^	30
	Tobramycin ^b^	10
	Gentamicin ^a^	10
	Netilmicin ^b^	30
	Neomycin ^a^	30
Folate pathway antagonists	Sulfonamides ^a^	300
	Trimethoprim–Sulfamethoxazole ^a^	23.75–1.25
Phosphonic acid derivatives and combinations	Fosfomycin ^c^	200
	Fosfomycin–Tylosin ^c^	160–40
Lipopeptides	Colistin ^a^	10
Macrolides	Azithromycin ^a^	15
	Erythromycin ^a^	15
Nitro-heterocyclics	Nitrofurantoin ^c^	300
Phenicols	Chloramphenicol ^a^	30
	Florfenicol ^a^	30
Quinolones and Fluoroquinolones	Nalidixic acid ^a^	30
	Ciprofloxacin ^a^	5
	Norfloxacin ^a^	10
	Ofloxacin ^b^	5
	Levofloxacin ^c^	5
	Enrofloxacin ^a^	5
Tetracyclines	Tetracycline ^a^	30
	Doxycycline ^a^	30
	Minocycline ^c^	30
	Tigecycline ^a^	15

^a^ Oxoid, Basingstoke, UK. ^b^ Liofilchem, Via Scozia, Italy. ^c^ Britania, Buenos Aires, Argentina.

**Table 3 animals-14-01207-t003:** Doses of commercial disinfectants used for antimicrobial susceptibility testing of *Salmonella* spp. strains.

Name of the Disinfectant (Company)	Ingredients	Manufacturer’s Dose Recommendation (%)	Tested Doses (%) from Original Product
PL301 AS (Proar Pilar S.A., Villa Bosch, Argentina)	Caustic soda, alkyl glucoside, polyalkali, sodium gluconate, water	3–5	0.03; 0.06; 0.12; 0.25; 0.5; 1; 2, 4; 8
PL308 DA (Proar Pilar S.A., Villa Martelli, Argentina)	Alkaline agents, non-ionic surfactants, ethylenediaminetetraacetic acid (EDTA), corrosion inhibitors, water	1	
Ucarsan 420 (Ruminal, Villa Martelli, Argentina)	Glutaraldehyde, detergent	0.5	
Ruminal 50 (Ruminal)	Alkyl dimethyl benzyl ammonium chloride	0.04	
End Bac 2 (Diversey, Villa Bosch, Argentina)	Sodium hypochlorite	0.55–2.5	
GLUTASEPT (Nieser, Argentina S.A., Pilar, Buenos Aires)	Formaldehyde, glutaraldehyde, alkyl dimethyl benzyl ammonium chloride, glyoxal	0.25–0.50	
Cid 20 (CID LINES, Ieper, Belgium)	Alkyl dimethyl benzyl ammonium chloride, glyoxal, glutaraldehyde, isopropanol, formaldehyde	0.25–0.50	
X5 (Laboratorios Bimex S.R.L., Ciudad Autonoma de Buenos Aires, Argentina)	Formaldehyde, dimethyl benzyl alkyl ammonium chloride, glutaraldehyde, glyoxal, isopropanol	0.5–2	
Squad (Nieser Argentina S.A., Pilar, Argentina)	Mixed alkyl dimethyl benzyl ammonium chlorides, formaldehyde, ethyl alcohol	1	
TH4+^®^ (Phibro Animal Health, Grand Bourg, Argentina)	Didecyl dimethyl ammonium, alkyl dimethyl benzyl ammonium chloride, octyl dicyldimethyl ammonium chloride, dioctyl dimethyl ammonium chloride, glutaraldehyde, pine oil, terpinol	0.2–0.5	
Zix Virox^®^ (Vetanco S.A., Villa Martelli, Argentina)	Hydrogen peroxide, peracetic acid	0.2–1.5	
El Rey Granel	Alcohol vinegar	pure	20; 30; 40; 50; 60; 70; 80; 90; pure
Pramolipa (Pramol Química, Piñeyro, Argentina)	Isopropyl alcohol	70	
Germatron (Quimax S.A., Ciudad Autonoma de Buenos Aires, Argentina)	Pine oil, quaternary ammonium	0.82–1	0.05; 0.1; 0.2; 0.4, 0.8; 1.6; 3.2; 6.4, 12.8
OmnicideTM (Ensol, Villa Bosch, Argentina)	Glutaraldehyde, dimethyl cocobenzyl ammonium chloride	0.2–0.3	
Profoam (Diversey, Villa Bosch, Argentina)	Potassium hydroxide, tripolyphosphate sodium, sodium hypochlorite, adjuvants	3–10	0.16; 0.31; 0.62; 1.25; 2.5; 5; 10; 20, 40
Aminol 50 (Laboratorios Weende, Villa Maria, Argentina)	N-alkyl dimethyl benzylammonium chloride	0.02–0.04	0.0025; 0.05; 0.1; 0.2; 0.4; 0.8; 1.6; 3.2; 6.4

**Table 4 animals-14-01207-t004:** Antibiotic non-susceptibility profiles of two strains of *Salmonella* ser. Golstrup isolated from a canary pool of cage papers. I = intermediate, R = resistant, S = susceptible.

Antibiotic	*Salmonella* ser. Golstrup	
	Strain 1	Strain 2
Cefazolin	I	I
Ceftaroline	I	S
Ciprofloxacin	I	I
Erythromycin	R	R
Neomycin	I	S
Sulfonamides	R	R

**Table 5 animals-14-01207-t005:** The minimum inhibitory dose (MID) of the disinfectants tested against two strains of *Salmonella* ser. Golstrup. Sensitive (S), intermediate (I), and resistant (R) when the MID is less than, equal to, and greater than the maximum dose recommended by the manufacturer, respectively.

Name of the Disinfectant	MID of the Disinfectants against *Salmonella* ser. Golstrup	
	Strain 1	Strain 2
PL301 AS	1 (S)	
PL308 DA	>8 (R)	
Ucarsan 420	8 (R)	1 (R)
Ruminal 50	0.12 (R)	<0.03(S)
End Bac 2	8 (R)	
GLUTASEPT	2 (R)	1 (R)
Cid 20	1 (R)	
X5	2 (I)	0.25 (S)
Squad	1 (I)	0.5 (S)
TH4+^®^	0.5 (I)	0.25 (S)
Zix Virox^®^	2 (R)	0.06 (S)
El Rey Granel	>pure (R)	60% (S)
Pramolipa	Pure (R)	80 (R)
Germatron	3.2 (R)	
Omnicide^TM^	0.8 (R)	0.2 (S)
Profoam	40 (R)	5 (S)
Aminol 50	0.08 (R)	

## Data Availability

The data used in the present work are presented in the tables.

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
