# Peer review of "Study of Salmonella spp. from Cage Papers Belonging to Pet Birds in an Argentinean Canary Breeder Championship"

_animals, 2024, doi:10.3390/ani14081207_

Round 1
Reviewer 1 Report
Comments and Suggestions for Authors
Authors,
Please use the attached to revise the manuscript.
Reviewer.

Comments on the Quality of English LanguageMust be improved.
Author Response
Thank you very much for taking the time to review this manuscript. Please find the detailed responses below and the corresponding revisions/corrections follow the “Track Changes” function in the revised manuscript. Most of the recommendations were incorporated in the revised manuscript. Other are explained below:
Page 3, line 123: 2.2. Salmonella isolation and identification were explained below in the text. We consider that it is not necessary to have any reference in this title because the technique used was not an exact copy from others.
Page 6, line 189: Two hundred fifty-eight sterile bags were the samples for Salmonella spp. isolation. It was clarified in page 3 line 117.
Page 7, line 190. EF-18 agar is used for the selective and differential isolation of Salmonella (https://www.neogen.com/globalassets/pim/assets/original/10001/6901_pi.pdf)
Page 7, line 206: Table 5. The minimum inhibitory dose (MID) Table 5. The minimum inhibitory dose (MID)… “The manufactures recommended doses and the doses tested for each disinfectant were in Table 3.
Page 8, lines 230-248. The referee suggested to eliminate these sentences for not relevant content. However, we consider that these sentences corresponded to the discussion of Salmonella spp., isolated from pet birds, antibiotic sensibility. So, we rewrote these sentences to clarify their utility, including the antibacterial sensitivity of Salmonella ser. Golstrup strains.
Page 8, line 264. We think that it is not necessary to include the number of tables in the discussion section, which is in the result section. We checked different papers from Animals, and we did not see that in the discussion section.
Page 9, line 279: We think that it is not necessary to include the number of tables in the conclusion section, which is in the result section. We checked different papers from Animals, and we did not see that in the conclusion section.
Reviewer 2 Report
Comments and Suggestions for Authors
The manuscript entitled “Study of Salmonella spp. from cage papers belonging to pet birds in an Argentinean Canary Breeder Championship” is well written and well structured in each of its sections and addresses a topic that may be of interest to many researchers.
Although the number of antibiotics tested is very large and thus accounts for the difficulty of being able to perform a micromethod MIC determination for all of them, at least for some, e.g., the most representative for each class, it would have been desirable as it would have represented a decidedly more quantitative and indisputable figure.
Realizing that the request for such an addition is excessive for the Authors who would be forced to set up a new and demanding package of laboratory tests, for which they may not even be equipped, I simply request that they justify the choice of mediate determination using the disk diffusion technique for their experimental plan.
I have no further comments or suggestions for the other sections of the manuscript. I congratulate the Authors for their excellent work and wish them all the best in seeing their research published soon.
Author Response
Thank you very much for taking the time to review this manuscript.
“I simply request that they justify the choice of mediate determination using the disk diffusion technique for their experimental plan.”
We chose the standard disk diffusion method because it is simple, reproducible and we had disks of many antibiotics to test. If you want, we can include these reasons in the text of the manuscript.
Reviewer 3 Report
Comments and Suggestions for Authors
This study phenotypically characterizes the antimicrobial and disinfectant profiles of Salmonella isolates recovered from pet birds in Argentina. I have the following comments for the authors:
Line 72: “anti-microbial” consider revising to “antimicrobial”
Line 147: “National Committee for Clinical Laboratory Standards (NCCLS) is now known as the Clinical and Laboratory Standards Institute (CLSI) revise accordingly.”
Provide information on the reference control strains that were used for antimicrobial susceptibility testing
Line 178-179: “was less, and equal than” consider revising to “was less than, and equal to”
Line 181: “181-183” This sentence is unclear. Please consider rewriting it to ensure proper grammar and readability
Line 196-197: “There were not any MDR strains” consider revising to “No MDR strains were identified”
Line 233: “Although most of strains” consider revising to “Although most of the strains”
Table 3: Doses of commercial disinfectants used for antimicrobial susceptibility test over Salmonella spp. Strains revise to “Doses of commercial disinfectants used for antimicrobial susceptibility testing of Salmonella spp. strains”. Consider revising the subheading “Manufacture’s doses recommendation (%)” to “Manufacturer’s recommended doses (%)”
I would suggest rounding the numbers under “Tested doses (%) from original product” to the closest decimal to ensure readability.
Comments on the Quality of English Language
The manuscript would benefit from proofreading to ensure quality and readability.
Author Response
Thank you very much for taking the time to review this manuscript. Please find the detailed responses below and the corresponding revisions/corrections follow the “Track Changes” function in the revised manuscript.
Line 72: “anti-microbial” consider revising to “antimicrobial”
Considered.
Line 147: “National Committee for Clinical Laboratory Standards (NCCLS) is now known as the Clinical and Laboratory Standards Institute (CLSI) revise accordingly.”
Changed.
Provide information on the reference control strains that were used for antimicrobial susceptibility testing
Escherichia coli ATCC 25922 was used as control strain.
Line 178-179: “was less, and equal than” consider revising to “was less than, and equal to”
Changed.
Line 181: “181-183” This sentence is unclear. Please consider rewriting it to ensure proper grammar and readability
Re-wrote.
Line 196-197: “There were not any MDR strains” consider revising to “No MDR strains were identified”
Changed.
Line 233: “Although most of strains” consider revising to “Although most of the strains”
Corrected.
Table 3: Doses of commercial disinfectants used for antimicrobial susceptibility test over Salmonella spp. Strains revise to “Doses of commercial disinfectants used for antimicrobial susceptibility testing of Salmonella spp. strains”. Consider revising the subheading “Manufacture’s doses recommendation (%)” to “Manufacturer’s recommended doses (%)”
Changed.
I would suggest rounding the numbers under “Tested doses (%) from original product” to the closest decimal to ensure readability.
Corrected.
Round 2
Reviewer 1 Report
Comments and Suggestions for Authors
Authors,
Please review and revise as recommended in the attached document.
Reviewer.

Comments on the Quality of English LanguageCan be improved.
Author Response
Thank you very much for taking the time to review this revised manuscript. Most of the recommendations were incorporated in the revised manuscript revisions/corrections follow the “Track Changes” function in the second version of the manuscript. Furthermore, although it was not originally requested, we added some clarification of breakpoints used for colistin in materials and methods section. Three suggestions were not incorporated, which were explained below:
- Results
Page 7, line 198 and Table 4: “Salmonella ser. Golstrup strains were not MDR strains and the MARI for these strains 198 was 0.04.” Because of these strains were resistant to sulfonamides and erythromycin, MARI was 2/52=0.04. It was added in the text and corresponded to table 4, which was written two sentences later, but the information of MARI was not incorporated in this table to avoid repetition of information.
- Conclusions
Page 9, line 279: The reviewer suggests to change the word “sample” by “effective sampling method”. “Cage paper” is a sample and it is not a “sampling method”. So, the name of the “sample” was corrected to singular form to clarify the sentence.